# EEG Signal Complexity Measurements to Enhance BCI-Based Stroke Patients’ Rehabilitation

**DOI:** 10.3390/s23083889

**Published:** 2023-04-11

**Authors:** Noor Kamal Al-Qazzaz, Alaa A. Aldoori, Sawal Hamid Bin Mohd Ali, Siti Anom Ahmad, Ahmed Kazem Mohammed, Mustafa Ibrahim Mohyee

**Affiliations:** 1Department of Biomedical Engineering, Al-Khwarizmi College of Engineering, University of Baghdad, Baghdad 47146, Iraq; 2Department of Electrical, Electronic and Systems Engineering, Faculty of Engineering and Built Environment, Universiti Kebangsaan Malaysia, UKM, Bangi 43600, Selangor, Malaysia; 3Centre of Advanced Electronic and Communication Engineering, Department of Electrical, Electronic and Systems Engineering, Universiti Kebangsaan Malaysia, UKM, Bangi 43600, Selangor, Malaysia; 4Department of Electrical and Electronic Engineering, Faculty of Engineering, Universiti Putra Malaysia, UPM, Serdang 43400, Selangor, Malaysia; 5Malaysian Research Institute of Ageing (MyAgeing)^TM^, University Putra Malaysia, Serdang 43400, Selangor, Malaysia

**Keywords:** brain computer interface, electroencephalogram, motor imagery, independent component analysis, features, entropy, Laplacian, classification

## Abstract

The second leading cause of death and one of the most common causes of disability in the world is stroke. Researchers have found that brain–computer interface (BCI) techniques can result in better stroke patient rehabilitation. This study used the proposed motor imagery (MI) framework to analyze the electroencephalogram (EEG) dataset from eight subjects in order to enhance the MI-based BCI systems for stroke patients. The preprocessing portion of the framework comprises the use of conventional filters and the independent component analysis (ICA) denoising approach. Fractal dimension (FD) and Hurst exponent (Hur) were then calculated as complexity features, and Tsallis entropy (TsEn) and dispersion entropy (DispEn) were assessed as irregularity parameters. The MI-based BCI features were then statistically retrieved from each participant using two-way analysis of variance (ANOVA) to demonstrate the individuals’ performances from four classes (left hand, right hand, foot, and tongue). The dimensionality reduction algorithm, Laplacian Eigenmap (LE), was used to enhance the MI-based BCI classification performance. Utilizing k-nearest neighbors (KNN), support vector machine (SVM), and random forest (RF) classifiers, the groups of post-stroke patients were ultimately determined. The findings show that LE with RF and KNN obtained 74.48% and 73.20% accuracy, respectively; therefore, the integrated set of the proposed features along with ICA denoising technique can exactly describe the proposed MI framework, which may be used to explore the four classes of MI-based BCI rehabilitation. This study will help clinicians, doctors, and technicians make a good rehabilitation program for people who have had a stroke.

## 1. Introduction

Common symptoms that have a substantial impact on the quality of life of stroke survivors include disability and cognitive impairment; stroke is the most frequent cause of disability and impairment [1]. Daily activities are significantly impacted by those impairments [2]. Recent studies have concentrated on developing effective therapies and rehabilitation programs for stroke victims. Helping the brain repair neuronal connections and make up for damaged circuits is the goal of therapy and rehabilitation. Though choosing the appropriate course of action can take weeks, it is still not yet possible to do so objectively [3].

Researchers have discovered in recent years that brain–computer interface (BCI) techniques can improve communication between the brain and computer by decoding brain neural signals [4,5]. There have been several attempts to investigate the application of BCI, but motor imagery (MI) has received the most attention since it causes the motor cortex to respond when a person mentally models a specific movement of limbs without activating their muscles [4]. Even in the presence of significant nerve injury, repetitive training based on MI encourages neuronal reorganization [6,7]. In conjunction with specific external assistive technologies, the MI-based system has the potential to significantly improve the quality of life for individuals with stroke, spinal cord injury, and amyotrophic lateral sclerosis [8,9].

Researchers used event-related desynchronization and event-related synchronization (ERD/ERS) to classify mental states [10,11], because the sensorimotor cortex attenuates oscillatory brain activity within specific frequency bands [12,13], which can be distinguished when the subject imagines moving different sides of the body [14,15]. EEG signals can be used to track a patient’s health and brain activity changes [16].

All prior research contributes significantly to the MI classification task. Classifiers such as support vector machines (SVM), linear discriminant analyses (LDA), and random forests (RF) have all been employed [5]. Nevertheless, most previous research only employs a single conventional filter during the preprocessing (denoising) stage, and most forecast MI EEG data analysis techniques overlook the benefits of using a hybrid filtering strategy for denoising the EEG signal before deploying algorithms. Most previous research has employed feature extraction approaches that are specific to a single domain, such as CSP or WT [17], during the feature extraction step. Existing research has not shown a variety of feature extraction and dimensionality reduction techniques to evaluate the MI EEG’s complexity and irregularity, an area where much opportunity exists to enhance classification accuracy.

Apart from other studies, in this work, the preprocessing stage was where the conventional filters and independent component analysis (ICA) denoising technique were initially applied. Then, nonlinear features were retrieved, such as fractal dimension (FD), Hurst exponent (Hur), and Tsallis entropy (TsEn), as well as dispersion entropy (DispEn), as dynamic entropy parameters. Two-way analysis of variance (ANOVA) was used to statistically analyze four MI-based classes. Due to the effectiveness of the used features, they were combined into CompEn integrated feature set. Laplacian Eigenmap (LE) dimensionality reduction algorithm was applied to the feature set to improve the classification performance of the motor imagery (MI)-based BCI stroke patients.

Not all of the time will a decision BCI MI-based EEG dataset system show promising outcomes; its advantages and disadvantages should be weighed carefully. In particular, unlike other methods that involve a great deal of training and expensive sensors, EEGs are a wireless, low-cost device that can automatically recognize MI from EEG data. When compared to magnetoencephalogram (MEG) systems, the proposed CompEn integrated feature set can investigate complexity and irregularity over the entire spectrum of MI-based EEG data with normal throughput and can be mastered by a greater number of users.

To the author’s knowledge, this work has addressed the contribution of the EEG role in the MI-based BCI interaction. This paper’s primary novelty, therefore, has two components. Its first objective is to suggest a wireless EEG system based on low-cost, automated EEG signals that can recognize MI. This can be accomplished by examining the complexity and irregularity over the entire spectrum of MI-based EEG data and proposing the CompEn integrated feature set. Second, the feature set was subjected to the Laplacian Eigenmap (LE) dimensionality reduction approach to improve classification performance in the development of the MI framework, which can be sensitive for detecting a four-MI-class (left hand, right hand, feet, and tongue) from eight patients.

## 2. Related Works

EEG is a non-invasive method for identifying conditions and symptoms that affect the brain. Numerous neurological conditions, including epilepsy, tumors, cerebrovascular lesions, depression, and trauma-related issues, can be diagnosed with its aid. An emerging method of direct brain-to-computer communication, known as EEG-based BCI, relies on the interpretation of features that are extracted from the EEG signal with higher resolution than those found in signals from other devices [18].

The EEG potentials recorded from the scalp’s surface are bioelectric signals produced by the neuronal activity of the brain. Many researchers have been concentrating on the fundamentals of BCI in recent years through the interpretation of EEG-based commands and have worked on controlling a device with this approach [18]. These research studies continue to contribute to raising the quality of life for those who are paralyzed or have lost limbs such as their arms and legs [19]. An EEG-based BCI system that Mabrouk et al. [20] have created shows the user’s EEG signals and accurately categorizes them as fictitious right and left hand movements.

According to studies conducted by [21], the active EEG frequency bands and brain regions that contribute to cognitive load fluctuate based on the learning state. Moreover, they reported variations in EEG frequency bands in specific brain regions under cognitive stress when performing activities involving human–computer interaction [22].

EEG is, therefore, trustworthy and among the most sensitive indicators of brain activities for determining mental burden brought on by cognitive processing [23].

Recognizing the most pronounced marks from EEG signals is essential to detecting and identifying brain features as well as assessing the EEG signal variable under evaluation [3]. From a clinical standpoint, the neurologist interprets the post-stroke patient’s EEG signal by looking at wave rhythms, amplitudes, asymmetries, changes in magnitudes, the presence of waves, and the ratio between waves [24,25].

The recorded wave activities could be distorted, though, by a variety of artifacts [26,27]. The abnormal behavior of the brain can typically be imitated or superimposed by these artifacts. Additionally, significant artifacts that conflict with EEG, such as eye blinks and ocular movements, cardiac artifacts, muscle activities, and noise from power lines, may cross EEG frequencies [28]. Thus, it is challenging to categorize EEG signals due to noise [29].

In order to improve our understanding of how the brain works, it is crucial to quantify the complexity of the EEG signal [30]. This allows us to obtain insight into the process and distinguishing characteristics of the signal. When examining the complexity changes caused by events in the functional areas of the brain, nonlinear parameters are very useful indicators [31]. It should be noted that nonlinear parameters are frequently employed for a range of neurophysiological investigations and applications utilizing EEG signals [32]. The EEG signal is difficult to acquire, process, and analyze due to its complexity and nonstationarity. When data processing for feature extraction is properly prioritized, significant information about the neurophysiological conditions in the brain can be gleaned.

Due to the numerous redundant data reductions and transformations involved in data processing, one must be extremely careful when selecting the best methodology to prevent information loss. Using advocates of nonlinear discrete dynamical systems, the authors of previously published studies have examined changes in the temporal dynamics of the EEG signal under moderate to demanding mental stimulation. However, it has been suggested that the brain in the majority of these cases functions as a nested network of coupled dynamical systems that maintains spatial and temporal dynamics and can be identified through nonlinear biomarkers of EEG signal [5,24].

The most recent techniques for EEG analysis for MI-based BCI are shown in Table 1, which also includes feature extraction and classification methods as well as various denoising techniques. Bandpass filters and notch filters have been used as denoising techniques in many studies, see [33,34,35], but blind source separation algorithms (BSS) have also been used with ICA in other studies (see [36,37,38]). As in [17,33,35,38], CSP was also the most widely used feature extraction technique. In reality, different EEG-based BCI datasets show differences in the brain regions associated with MI-BCI, making it difficult for conventional methods of feature extraction to demonstrate accurate classification of different classes.

As far as the authors of this article are aware, the majority of studies computed the nonlinear dynamical parameters. The complexity and irregularity characteristics used in this study, however, may aid in understanding how specific spatial information of brain functions changes over time [39]. The majority of EEG-BCI-based motor imagery studies published in the literature concentrate on separating left from right hand or foot motor imagery [19].

As a result, this issue could be resolved by using more effective features that are compatible with the complexity of the brain and that can be used to elicit the unique performance of subjects following motor imagery (MI)-based BCI rehabilitation.

## 3. Methods and Materials

This study’s methodology entails examining nonlinear MI-BCI EEG-based time series in an effort to find variations in complexity and entropy occurring during MI-BCI. Figure 1 shows the directions for the MI-based BCI framework, including the stages of preprocessing, feature extraction, statistical analysis, LE dimensionality reduction, and classification.

### 3.1. Data Description

The dataset used is BCI Competition IV dataset 2a, [22], which is a four-class (left hand, right hand, foot, and tongue) MI dataset from eight subjects donated by Graz University (Table 2), homepage (http://www.bbci.de/competition/iV accessed on 12 January 2023). To collect the dataset, 22 EEG channels and 3 EOG channels are employed at a sampling rate of 250 Hz. The original EEG with 22 channels is shown in Figure 2.

Each subject’s data consists of two sessions: a training session and a testing session. There are a total of 288 trials for the four classes in each session, with an average of 72 trials per class. Throughout the data collection process, 22 Ag/AgCl electrodes were used, and the amplifier’s sensitivity was adjusted to 100 V. The 8–32 Hz band is used to filter the raw EEG signal [23]. Because the alpha and beta frequency bands, which have been found to be the most pertinent for MI categorization, are included in the frequency band, it was chosen. BCI Competition IV [22] contains a thorough overview of the dataset.

A cross and a brief warning tone are displayed on the blank screen at the start of each trial (t = 0 s). An arrow that points left, right, below, or above shows on the screen two seconds later (t = 2 s) and remains there for four seconds. Up until the prompt arrow vanished from the screen (t = 6 s), the subject completed the necessary motor imaging tasks. Following a brief intermission, the screen became completely black for the following motion imagination experiment (Figure 3) [4].

### 3.2. Preprocessing Stage

Because the MI-based BCI was contaminated by so many different artifacts (Figure 4), conventional filtering and the independent component analysis (ICA) techniques have been used as described in the following sections.

#### 3.2.1. Conventional Filtering

To obtain the EEG dataset, 22 scalp-area channels were used. Each channel of the acquired EEG datasets was processed individually using conventional filtering techniques in the first stage. The power line interference noise was eliminated using a notch filter at 50 Hz, and the band of the recorded EEG signals was limited using a band pass filter (BPF) with a frequency around (8–30) Hz.

#### 3.2.2. Independent Component Analysis (ICA)

Therefore, the ICA technique has been utilized to identify and eliminate various artifacts from EEG data. First, the ICA algorithm and the linear mixing model were used since ICA is a potent statistical technique for separating mixed signals based on a number of presumptions. The most crucial premise is that the sources already in existence be statistically independent of one another, and the mixing procedure should be instantaneous and linear [42,43].


The goal of ICA is to estimate the set of n unknown components, $s\left(t\right)=\left[s_{1}\left(t\right),\ldots ,\;s_{n}\left(t\right)\right]$, which were linearly mixed by the matrix A, x(t) is the set of observations and $x\left(t\right)=\left[x_{1}\left(t\right),\ldots ,\;x_{n}\left(t\right)\right]$ [42,44]. Where x(t) represents the EEGs, which are related to s(t), t is the time or sample index, the ICA linear transform Equation is:(1)x(t)=As(t)
where x(t) and s(t) are supposed to have zero mean. The ICA uses the higher-order statistics of x(t) to compute the demixing matrix W, which is the inverse matrix of A, to be linearly represented the independent components. Then, under such assumptions, the ICs can be estimated by Equation (2) [42,44,45]:(2)y(t)=Wx(t)
where $y\left(t\right)=\left[y_{1}\left(t\right),\ldots ,\;y_{n}\left(t\right)\right]$ is the vector that estimate the ICs.

As a result, ICA is regarded as an effective method for identifying artifact components and brain activity components that may be affected more than other components [46,47,48].

In this study, denoising the EEG signals to increase characteristics during MI tasks was the main objective rather than isolating specific physiological activity. It is necessary to define a standard to compare ICs from various EEG epochs and participants and to identify which components are noise-sensitive [46,47,48]. This criterion was met by decomposing EEG data using the SOBI method, which was based on SOBI algorithm [49,50]. In order to extract the new component matrix $\;\bar{S}\;$ and modify for various circumstances, the SOBI algorithm was chosen because of its robustness, speed of convergence, and minimal number of tunable parameters [51].

The ICA method is used to remove artifacts from EEG recordings; however, its implementation depends heavily on the user, because one of ICA’s biggest drawbacks is that it cannot identify the estimated origins. The application of the artifact detection metrics based on the estimation of cross correlation (XCorr) overcame this restriction. The data will be cleared of all artifactual ICs marked using XCorr (zeroing the artifactual ICs method). Equation (3) gives the correlation XCorr between the interest-related EEG signal x and the EOG noisy signal y.
(3)$$XCorr\left(x,y\right)=\frac{\sum \left(x-\bar{x}\right)\left(y-\bar{y}\right)}{\sqrt{\sum {\left(x-\bar{x}\right)}^{2}{\left(y-\bar{y}\right)}^{2}}}$$

The original, artifact-free EEG data will be used to estimate the ICA for the artifactual free ICs, which will be reconfigured as the new dataset. As stated in Equation (4), the ICA calculated of the original, artifact-free EEG data was reconstructed from the ICA corrected EEG signals to create $\hat{x}$, the new data set, where $\hat{s}\left(t\right)$ is the new component matrix.
(4)$$\hat{x}\left(t\right)=A\hat{s}\left(t\right)$$

### 3.3. Features Extraction

#### 3.3.1. Complexity Analysis

The EEG signal is the key to determining the complexity of the nonlinear activity of the brain, which is why the human brain is regarded as a dynamical system. The EEG signal’s dynamical system state can be determined by calculating the signal’s complexity [49]. The FD and Hur nonlinear time series approaches [50] are extremely effective and accurate in describing the MI-based EEG times series [51], which is how physiological signal complexity is best evaluated.

##### Fractal Dimension (FD)

Higuchi’s fractal dimension FD is a suitable technique for assessing biomedical signals [52] due to its reliance on a binary sequence and, frequently, decreased sensitivity to noise [53]. Since MI-based BCI rehabilitation uses EEG inputs, this work employed FD to look at such signals. The procedure to compute the HFD can be explained as follows:

Given a one dimensional time series X=x[1],x[2],…, x[N], the algorithm to compute the HFD can be described as follows [54]:(5)$$X_{k}^{m}=\left\{x\left[m\right],\;x\left[m+k\right],x\left[m+2k\right],\ldots ,x\left[m+\lfloor \frac{N-m}{k}\rfloor \times k\right]\right\}$$
where k and m are integers, k indicates the discrete time interval between points, whereas m=1,2,…,k represents the initial time value. For each of the k time series $X_{k}^{m}$, the length $L_{m}\left(k\right)$ can be computed as in Equation (6):(6)$$L_{m}\left(k\right)=\frac{\left\{\left(\sum_{i=1}^{\left\lfloor \frac{N-m}{k}\right\rfloor }\left|x\left[m+ik\right]-x\left[m+\left(i-1\right)\times k\right]\right|\right)\frac{\left(N-1\right)}{\lfloor \frac{N-m}{k}\rfloor \times k}\right\}}{k}$$
where N is the length of the original time series X, and the term $\frac{\left(N-1\right)}{\lfloor \frac{N-m}{k}\rfloor \times k}$ represents the normalization factor. Then, the length of the curve for the time interval k is defined as the average of the k values $L_{m}\left(k\right)$, for m=1,2,…,k:(7)$$L\left(k\right)=\frac{1}{k}\times \overset{k}{\underset{m=1}{\sum }}L_{m}\left(k\right)$$

Finally, the data should follow a straight line with a slope equal to the HFD of X when L(k) is plotted against 1/k on a double logarithmic scale, with $k=1,2,\ldots ,k_{max}$. Therefore, the slope of the line that least-squares fits the pairings {ln[L(k)], ln(1/k)} is the definition of HFD. HFD results were displayed against a range of $k_{max}$ in order to select an appropriate value for the parameter $k_{max}$. The HFD plateauing point is regarded as a saturation point, and that $k_{max}$ value ought to be chosen [55,56]. The greatest outcomes in this study were found while measuring the HFD of the EEG; $k_{max}=20$. HFD was computed using 6-s windows (1500 samples).

##### Hurst Exponent (Hur)

Numerous studies have looked at using the Hur to analyze EEG information from MI-based BCI rehabilitation [14]. The Hur is a metric for a time series’ long-memory characteristics [15,16]. The scalar depicts a time series’ relative propensity to either substantially regress to the mean (mean-reverting pattern) or cluster in a certain direction (trending pattern).

The Hur of time series, which serves as a gauge of long-range correlation, can be estimated using the R/S approach [17]. The following can be used to describe this technique.

Calculate the logarithmic retunes of detrended time series with length N=r−1, where t has length of original time series.
(8)$$N_{i}=\log\left(t_{i}+\frac{1}{t_{i}}\right)i=1,2\ldots .r-1$$Split the time series into m adjoining subsets $S_{j}$ of length n, where m×n=N, and j=1,2,…,m. The segments of each subset calls $N_{kj}$, with k=1,2,…,n.
The average of each subset $S_{j}$ is counted by:(9)$$M_{j}=\frac{1}{n}\Sigma_{k1}^{n}\Sigma_{j}^{m}N_{kj}$$Calculate the addition of deviation from the average for each subset of $S_{j}$ as:(10)$$X_{kj}=\Sigma_{i}{}_{=1}^{k}\Sigma_{j}{}_{=1}^{m}\left(N_{ij}-M_{j}\right),k=1,2\ldots .n$$The mean relative range of any single subset is calculated as:

(11)
$$R_{I_{j}}=max\left(X_{k}{,}_{j}\right)-min\left(X_{k}{,}_{j}\right),1<k<n$$

In this step, standard deviation of each subgroup is considered:

(12)
$$S_{I_{j}}=\mathrm{sqrt}(\frac{1}{n}\Sigma_{k_{=1}}^{n}\Sigma_{j_{=1}}^{m}{\left(N_{kj}-M_{j}\right)}^{2}$$

The range $R_{I_{j}}$ of each subset is rescaled by the related standard deviation $S_{I_{j}}$. Therefore, the average $\frac{R}{S}$ measures for each window with length *n* is:(13)$${\left(\frac{R}{S}\right)}_{n}=\frac{1}{m}\Sigma_{j_{=1}}^{m}\left(\frac{R_{I_{j}}}{S_{I_{j}}}\right)$$All above steps should be repeated for different time periods.Plot $\log\left(\frac{R}{S}\right)n$ versus log(n): The slope of this graph shows the Hur [18].Hur values could be calculated using a rescaled range formula estimated by the above steps.

(14)RR=(22H−1−1)×n^Hur
where n is the number of data points and Hur is the average of all EEG signals [19,20,21]. While EEG signals exhibit considerable non-stationary properties, the $\frac{R}{S}$ technique only reliably calculates Hur for stationary time series [22]. Therefore, the non-stationarity of data issue needs to be addressed in order to examine the dynamical Hur of EEG signals [23]. To achieve this, processing the data inside a window size of 6 s is one option. This window size is big enough for the data to statistically behave like a stationary time series.

#### 3.3.2. Entropy Analysis

Entropy is the rate of information generation in dynamical systems. Entropy [24] is a metric describing the unpredictable nature of information content. Entropy estimate is based on the idea of estimating the level of randomness in a time series. The usefulness of entropy-based characteristics for categorizing emotional-based EEG signal has been established in few investigations [24]. There are various methods for estimating entropy, including sample, permutation, and approximate methods. TsEn and DispEn were used for this investigation because they are reliable, effective, and computationally efficient parametric methods for directly estimating entropy from the time series of an EEG signal [57,58].

##### 
Tsallis Entropy (TsEn)


The estimation of the EEG changes using time domain-dependent entropy has been shown extensively using TsEn entropies. For instance, TsEn was developed from the EEG of traumatic brain injury patients [17]. In this study, four classes of EEG signals from MI-based BCI rehabilitation were classified using TsEn. In this essay, we used Tsallis et al.’s definition of entropy to measure a signal’s degree of uncertainty [57].
(15)ST=∑i=1Np(xi)lnq(1p(xi))=1−∑pq(xi)q−1 
where xi are information events, p(xi) are the probabilities of xi, and the q-logarithm function is defined as:(16)lnq(x)=x1−q−11−q ∀x>0, q∈ℝ

Particularly, ${ln}_{1}\left(x\right)=ln\left(x\right)$. TsEn is a better option for examining the entropy of a system for which it may not be accurate to assume intense qualities, because, while Shannon entropy has intensive properties, it is generally extensive [21]. The non-extensivity of the system being measured is quantified by the parameter q. Given the link shown in Equation (12), if the likelihood that the information events within the signal will occur is known, the quantity of information contained inside an EEG signal, or the signal’s entropy, can be approximated. To assess the likelihood of the information events in EEG recordings, examples of known physical information events observed in other natural signals can be employed. In particular, crucial spots (local peaks, minima, and discontinuities) seem to distinguish information events in natural signals [59]. Sneddon [21] offered the following approximation of Tsallis entropy for q=2, based on the observation that local critical points seem to designate information in natural signals:(17)$$S_{T}\approx S_{T}=1-\frac{\frac{1}{N}\sum s_{i}^{2}}{\sigma^{2}}\;$$
where N is the total number of bins formed by the signal’s local critical points, $s_{i}^{2}$ is the variance inside each of these bins, and $\sigma^{2}$ is the overall signal variance. The discrete derivative $\left(X_{i+1}-X_{i}\right)/\Delta t$ was defined as having critical points at locations where it was equal to zero or changed sign from its initial value. Due to the relatively little length of the chosen EEG intervals utilized in the analyses reported, Sneddon’s approximation for Tsallis entropy was adopted [60].

##### 
Dispersion Entropy (DispEn)


A new irregularity indicator known as DispEn was proposed by Rostaghi et al. [58] and is based on symbolic dynamics or patterns. Data are transformed into a new signal with just a few different patterns, and the analysis of dynamic time series is reduced to a distribution of symbol sequences. Its purpose was to address the drawbacks of other entropy measures such as sample entropy and permutation entropy. As a result, unlike other entropy measures, DispEn can distinguish between various biological and mechanical states and is sensitive to changes in simultaneous frequency and amplitude values [58]. The length m of templates, the number of classes c that determine the number of patterns or classes to be taken into account in the computation, and the time delay d are the three internal parameters that dictate how DispEn is calculated. For the latter parameter, it is advised to use the value d=1, and for m and c, use cm<N; N is the length of the time series. When c is too high, slight variations in the signal might induce a change in class, making it susceptible to noise. When c is too low, always with c>1, signal values are too widely apart and lead to being assigned to the same class [58].

DispEn has been applied to a univariate signal $X=x_{1},x_{2},\;\ldots ,\;x_{n}$ with length N to perform the DispEn algorithm. Firstly, $x_{j}\left(j=1,\;2,\ldots ,N\right)$ are mapped to c classes with integer indices from 1 to c. The classified signal is $u_{j}\left(j=1,2,\ldots ,N\right)$ to compute the time series $u_{i}^{m,c}$ of an embedding dimension m and time delay d. Each $u_{i}^{m,c}$ is mapped to a dispersion pattern $\pi_{v_{0}v_{1}\ldots v_{m-1}}$. The number of possible dispersion patterns assigned to each vector $u_{i}^{m,c}$ is equal to $c^{m}$, since the signal has m members and each member can be one of the integers from 1 to c. Therefore, for each of $c^{m}$ potential dispersion patterns $\pi_{v_{0}v_{1}\ldots v_{m-1}}$, relative frequency is obtained as follows:(18)$$p\left(\pi_{v_{0}v_{1}\ldots v_{m-1}}\right)=\frac{\#\left\{i|i\leq N-\left(m-1\right)d,u_{i}^{m,c}\;has\;type\;\pi_{v0vm}\ldots_{vm}{}_{-1}\;\right\}}{N-\left(m-1\right)d}$$
where # means cardinality. In fact, $p\left(\pi_{v_{0}v_{1}\ldots v_{m-1}}\right)$ shows the number of dispersion patterns of $\pi_{v_{0}v_{1}\ldots v_{m-1}}$ that is assigned to $u_{i}^{m,c}$, divided by the total number of embedded signals with embedding dimension m. Finally, based on Shannon’s entropy, the DispEn can be calculated as in Equation (19) [58,61].
(19)$$DispEn\left(X,m,c,t\right)=-\Sigma_{\pi 1}^{c}p\left(\pi_{v0vm}\ldots_{vm}{}_{-1}\right)\times \ln(p\left(\pi_{v0vm}\ldots_{vm}{}_{-1}\right)$$

### 3.4. Statistical Significance Analysis

To determine the level of statistical significance of the MI-based BCI EEG dataset, two-way ANOVA tests were run four times. Following that, the Kolmogorov–Smirnov test was used to determine normality, and the Levene’s test was used to determine homoscedasticity. Using Duncan’s test, the post-hoc comparison was carried out. ANOVAs in SPSS 22 were used for the statistical analysis. The findings of the TsEn and DispEn irregularity parameters and the FD and Hur as complexity features were compared to see if there was a statistically significant difference at the 0.05 (0.95%) confidence level.

A two-way ANOVA on the FD traits was performed during the first session of the ANOVA. In this analysis, the nonlinear features FD was the dependent variable, and the group factor (the individual MI-BCI performance from four classes (left hand, right hand, feet, and tongue)) among the eight subjects was the independent variable. All statistical tests had a significance level of p<0.05.

On the Hur characteristics, a two-way ANOVA was conducted during the second ANOVA session. In this analysis, the nonlinear complexity Hur feature was the dependent variable, and the group factor (individual MI-BCI performance from four classes (left hand, right hand, feet, and tongue)) among the eight subjects was the independent variable. All statistical tests had a significance level of p<0.05.

Two-way ANOVA was conducted on the TsEn characteristics in the third ANOVA session. In this analysis, the nonlinear characteristics TsEn served as the dependent variable, while the group factor (the individual MI-BCI performance from four classes (left hand, right hand, feet, and tongue)) among the eight individuals served as the independent variables. All statistical tests had a significance level of p<0.05.

On the DispEn characteristics, a two-way ANOVA was performed during the fourth ANOVA session. In this research, the nonlinear irregularity features DispEn were the dependent variable, while the group factor (the individual MI-BCI performance from four classes (left hand, right hand, feet, and tongue)) among the eight individuals served as the independent variables. All statistical tests had a significance level of p<0.05.

### 3.5. Laplacian Eigenmap (LE) Dimensionality Reduction Algorithm

Belkin et al. [62] suggested LE to identify the projection that respects the intrinsic geometrical structure from all the data points and also consists of labels because LE [62] has multi-class issues. In order to identify regional organization in the data, it creates a nearest neighbor graph. The graph’s vertex points represent the data’s points, and its edges represent the connections between those points’ neighborhoods. The similarity between neighboring points is represented by the edges’ non-negative weights. LE calculates eigenvalues and eigenvectors for the generalized eigenvector problem given the similarity matrix W:(20)Ly=λDy 
where L=D−W is the graph Laplacian, and D is the diagonal weight matrix with D $D_{ii}=\sum jW_{ji}$ [62]. Let the first r smallest eigenvectors of the preceding equation be $y_{1},\ldots ,y_{r}$. The ith row of $Y=[y_{1},\ldots ,y_{r}]$ provides the new coordinate for point i. LE makes an effort to map related points as precisely as feasible. The objective function of LE is:(21)$$Y_{opt}=\underset{Y}{argmin}\underset{ij}{\sum }∥Y_{i}-Y_{j}∥{}^{2}W_{ij}=tr\left(Y^{T}LY\right)$$
with the constraint
(22)$$Y^{T}DY=1$$

### 3.6. Classification Stage

EEG data were thoroughly examined to identify the four classes: left hand, right hand, feet, and tongue among the eight subjects. Indeed, the accuracy of the classification results can be influenced by both the proper selection of dimensionality reduction techniques and the type of classifier. Three widely used classification methods for brain illnesses were used in this study: SVM, KNN, and RF.

The SVM classifier achieved its best performance using ten-fold cross-validation to optimize the complexity parameter C with values in the range of $-4\leq \log_{10}\left(C\right)\leq 4\;$in C values C∈ {0.0001, 0.001, 0.01, 0.1, 0, 10, 100, 1000, 10,000} on the training set. The greatest results for C values were obtained during testing when C was equal to 10. Multi-class SVM classifiers were developed using the radial basis function (RBF) kernel as their foundation. Additionally, the minimal misclassification rate from the training dataset was calculated to help choose the smoothing value for SVM training. The best value can only be found by methodically adjusting across several training sessions. As a result, the value in this study was changed between 0.1 and 1 at intervals of 0.1. The lowest misclassification rate was found to correspond to a value of σ=0.5 [26].

The KNN classifier requires the specification of the parameter *k*. At 2-point intervals, the value of *k* was changed between 1 and 9. In order to train the classifier to find the ideal value of *k*, the value of *k* = 5 was chosen arbitrarily. The Euclidean distance was calculated as a measure of similarity for classifying each trial using KNN.

The RF classifier is an ensemble approach that predicts individual trees using a cluster of decision trees during training [63]. Instead of utilizing a Gini index and information gain, random forests selects the root node and partition the features at random. The classifier’s output is implemented depending on the majority of votes from trees. Random forests is an extension of bagging that is excellent at classifying motor imagery tasks based on EEG signals [64].

The accuracy of this dataset’s learning based on 10-fold cross-validation was represented by taking the average of all of these accuracies.

## 4. Results and Discussion

The MI-BCI EEG datasets would be examined, and the outcomes for each subject would be calculated using statistical analysis and classification methods in terms of accuracy and confusion matrix. The findings are as given in the following sections.

### 4.1. Results of Preprocessing Stage

Due to the wide diversity of EEG aberrations, it was possible to successfully suppress (red color) the original noisy (blue color) EEG signals by employing conventional filters and the ICA technique for each EEG channel separately (blue color). Figure 5 depicts the Midline Frontal (Fz channel) prior to and following use of the denoising technique.

### 4.2. Results of Features Extraction

The distribution and features of the four classes can be better understood with the aid of the features’ visualization presented below, which was taken from the EEG MI-based BCI dataset (left hand, right hand, foot, and tongue).

#### 4.2.1. Complexity Analysis

The complexity features collected from the EEG MI-based BCI dataset are represented as a box plot for easy analysis. Figure 6 depicts the distribution of FD and Hur values, respectively, for the left hand, right hand, foot, and tongue for Subject 1, with the median FD and Hur values depicted by the innermost line in each box. Whiskers, the vertical lines on top and bottom of the boxes, can be used to gauge the degree of skew.

#### 4.2.2. Entropy Analysis

Visualizing the entropy features that were derived from the EEG MI-based BCI dataset can be performed with the use of a box plot. Figure 7 contains a rectangular box that illustrates the distribution of TsEn and DispEn values for the left hand, right hand, foot, and tongue for Subject 1. The line that is contained within the rectangular box illustrates the median value for TsEn and DispEn, respectively. Whiskers are the lines that run vertically along the top and bottom of these boxes. They are a handy tool for determining how much the skewness has changed.

### 4.3. Results of Statistical Significance Analysis

ANOVA has been used to quantify significant EEG changes; Hur has been performed to check the significant difference based on Hur among all EEG channels, thus, to discriminate among the four MI-BCI rehabilitation tasks. Table 3 presents a comparative plot of Hur, which is sufficient to detect important patterns and to estimate the MI-BCI performance from four classes (left hand, right hand, feet, and tongue) of eight subjects individually.

Hur performance (p < 0.05) can reveal an overview of the activities from each subject showing significant differences in left hand and tongue values for Subjects 1, 2, and 6 ($LH\_T_{S1,2,6}$). Moreover, the significant differences in foot values were obtained for Subjects 3 and 5 $\left(F_{S3,5}\right)$. For Subject 7, the significant differences were shown in right hand values ($RH_{S7}$), and for Subject 8, effects were shown in tongue ($T_{S8}$) during MI-BCI rehabilitation tasks.

ANOVA has been used to quantify significant EEG changes, FD has been performed to check the significant difference based on FD among all EEG channels, thus, to reveal the discrimination among four MI-BCI rehabilitation tasks. Table 4 presents a comparative plot of FD which is sufficient to detect important patterns and to estimate the MI-BCI performance from four classes (left hand, right hand, feet, and tongue) of eight subjects individually.

FD performance (p<0.05) can provide an overview of the activities from each subject showing significant differences in tongue values for Subjects 1 and 5 ($T_{S1,5}$) and in right hand values for Subjects 2 and 8 $\left(↑F_{S2,8}\right)$. Further inspection of the results showed a decrease in tongue values for Subject 6 ($RH_{S6}$). However, for Subject 2 (LH<T<F<RH), whereas for Subject 3 (RH<LH<T<F). For Subject 7 (LH<F<RH<T), for Subject 8 (F<RH<LH<T), and for Subject 9 (RH<T<F<LH) during MI-BCI rehabilitation tasks.

In order to reveal and distinguish subjects across the four MI-BCI rehabilitation tasks, ANOVA has been utilized to measure important EEG changes. TsEn has also been performed to assess the significant difference based on TsEn among all EEG channels. A comparison plot of TsEn is shown in Table 5 that is enough for identifying significant trends and estimating the MI-BCI performance from four classes (left hand, right hand, feet, and tongue) of eight subjects separately.

TsEn performance (p<0.05) can reveal an overview of the activities from each subject showing significant differences in left-hand values for Subjects 1, 6, and 9 ($LH_{S1,6,9}$). Another significant difference can be obtained in foot values for Subjects 2, 3, and 8 $\left(F_{S2,3,8}\right)$. However, there were significant differences in tongue values for Subject 7 ($T_{S7}$) and in feet and right-hand values for Subject 5 $\left(RH\_F_{S5}\right)$.

ANOVA has been used to quantify significant EEG changes; DispEn has been performed to check the significant difference based on DispEn among all EEG channels in order to provide a valuable marker among four MI-BCI rehabilitation tasks. Table 6 shows a comparative plot of DispEn, which is sufficient to estimate the MI-BCI performance from four classes (left hand, right hand, feet, and tongue) of eight subjects individually.

DispEn performance (p<0.05) can provide an overview of the activities from each subject showing significant differences in right-hand values for Subjects 1 and 5 ($RH_{S1,5}$); however, Subjects 1 and 2 could be differentiated through left hand and foot, respectively. Further inspection of the results showed significant differences in feet values for Subjects 2, 7, and 8 ($F_{S2,7,8}$), and significant differences for Subjects 3 and 9 in left-hand values ($LH_{S3,9}$); whereas, for Subject 9, significant differences in tongue ($T_{S9}$) can be obtained during MI-BCI rehabilitation tasks.

### 4.4. Results of Classification Stage

FD and Hur complexity features and TsEn and DispEn irregularity features were combined into the CompEn integrated set due to effectiveness from the previous stage. Then, SVM, KNN, and RF were the three classifiers utilized to evaluate the performance of the CompEn integrated set of features.

The findings displayed in Table 7 clearly demonstrate that the KNN and RF classifiers were superior to SVM, with RF scoring slightly higher than KNN. Subject 5 has the highest accuracy overall, with a score of 87.27%, indicating a higher possibility of recovering from a stroke.

KNN and RF classifiers provided the highest accuracy for Subject 1. SVM provided acceptable accuracy for the tongue class but inadequate accuracy for the other classes (Figure 8).

For Subject 2, KNN and RF have similar results. With SVM only giving a good to moderate accuracy for RA, the best accuracy for LA is KNN, while RA, foot, and tongue is RF (Figure 9).

For Subject 3, the best results were the RF and KNN, respectively. While SVM gave faulty results except for RA, the best accuracy for the foot is KNN, while LA, RA, and tongue are RF (Figure 10).

For Subject 5, RF and KNN have the best accuracy, while SVM only has good accuracy for RA. However, KNN has the best accuracy for RA, while RF has the best accuracy for LA, foot, and tongue (Figure 11).

The results for Subject 6 are good for RF and KNN, but poor for all classes for SVM. While KNN has good accuracy in LA, RF provided the greatest results for the RA, foot, and tongue (Figure 12).

Although tongue received acceptable accuracy from all three classifiers, LA, RA, and foot received poor accuracy from all three, with SVM receiving the best accuracy. The most accurate technology overall is still RF (Figure 13).

KNN has generally good accuracy with foot and tongue for Subject 8. SVM only has good accuracy in foot, whereas RF also achieves acceptable results (Figure 14).

For subject 9, KNN and RF provided moderate and comparable accuracy, whereas SVM provided inadequate accuracy (Figure 15).

## 5. Conclusions

This investigation endeavors to encourage the way toward characterizing the emotional EEG based on MI-based BCI rehabilitation from the basic EEG channels. Twenty-two channels were utilized to record the EEG signs of four classes (left hand, right hand, feet, and tongue). In the present study, ICA has been used as a denoising technique, complexity features using FD and Hur, and irregularity parameters including TsEn and DispEn were computed from each EEG channel. Two-way ANOVA has been performed to characterize the MI-based BCI rehabilitation performance. LE dimensionality reduction algorithm was used to reduce the CompEn integrated feature set dimension. SVM, KNN, and RF classification algorithms were applied to enhance the performance of the MI-based BCI systems for stroke patients using EEG signal processing. There are several limitations on this study that need to be made clear. The sample size was small, to start with. Additionally, the patients’ follow-up was not extensive. In addition, longer series with follow-up are required. Despite these limitations, all of our findings agree with those of other researchers who have made useful discoveries. RF was the best with 74.48% average accuracy of multiple performance measures, obtaining good classification results for all subjects. However, KNN classifier exhibits better results with 73.20% average accuracy compared to the SVM. Therefore, using an LE dimensionality reduction method has many benefits, including simplified computation complexity and enhanced feature extraction. Consequently, the proposed framework utilizing ICA denoising method, complexity, and irregularity features, classifiers, particularly the RF technique, were a crucial role in enhancing BCI-based stroke patients’ rehabilitation. Therefore, results suggest that BCIs can be a helpful tool in the treatment of motor deficits. For those with a spinal cord injury, a BCI-controlled neuroprosthesis can restore the use of their upper extremities; for those who have suffered a stroke, a BCI-triggered visual or neuroprosthesis based on functional electrical stimulation (FES) can encourage their own recovery. In order to move forward with widespread implementation of BCI-controlled neuroprostheses, it must be established beyond reasonable doubt that the experimental results obtained in a controlled laboratory setting can be reliably reproduced under real-world conditions with minimal loss of performance and stability. There are tasks related to this research that need to be improved and can be further applied to obtain better identification of motor imagery (MI) tasks. Some ongoing and future directions of extending the research are outlined as: firstly, channel selection can be applied to point the most efficient electrodes. Secondly, another possible research direction is the application of the empirical mode decomposition method (EMD) as a denoising technique. Thirdly, other classifiers using deep learning methods can be considered to compare the classification performance.

## Figures and Tables

**Figure 1 sensors-23-03889-f001:**
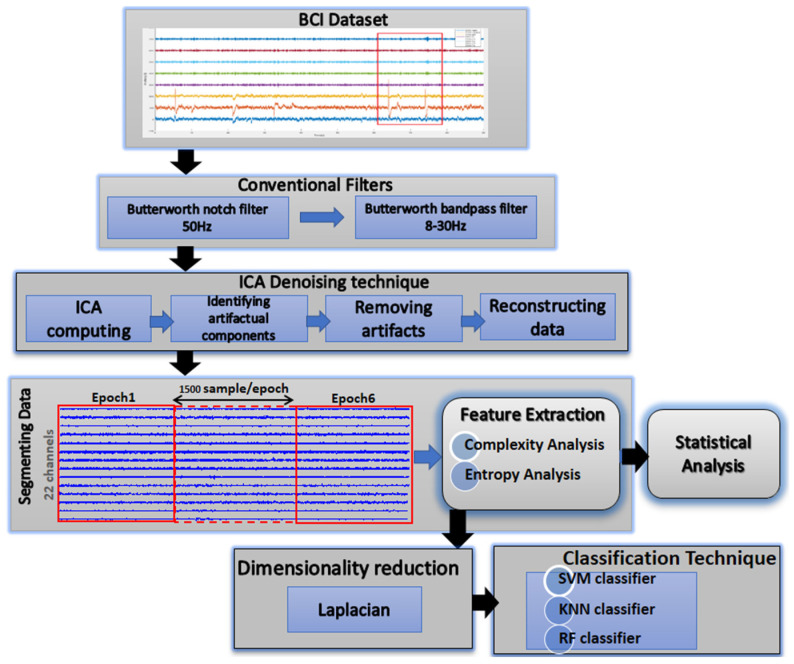
The suggested study’s block diagram.

**Figure 2 sensors-23-03889-f002:**
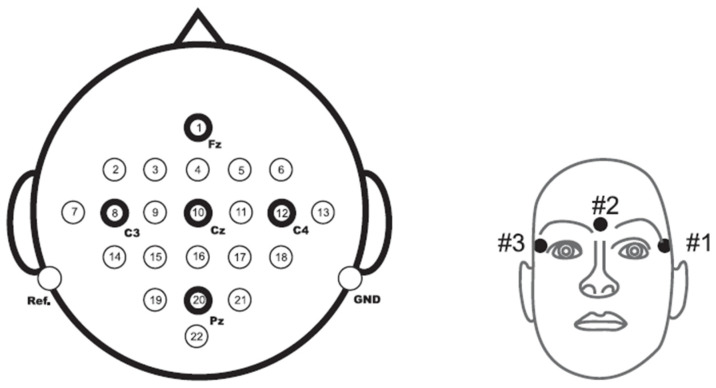
Distribution of EEG electrodes using the 10–20 system on the left and insertion of three monopolar EOG electrodes on the right [40].

**Figure 3 sensors-23-03889-f003:**
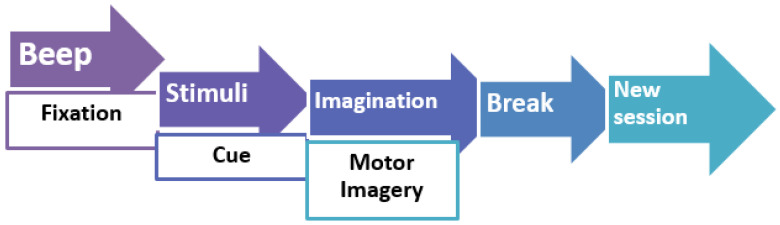
From BCI Competition IV dataset 2a, one trial timing scheme is provided for the MI experimental paradigm based on EEG [41].

**Figure 4 sensors-23-03889-f004:**
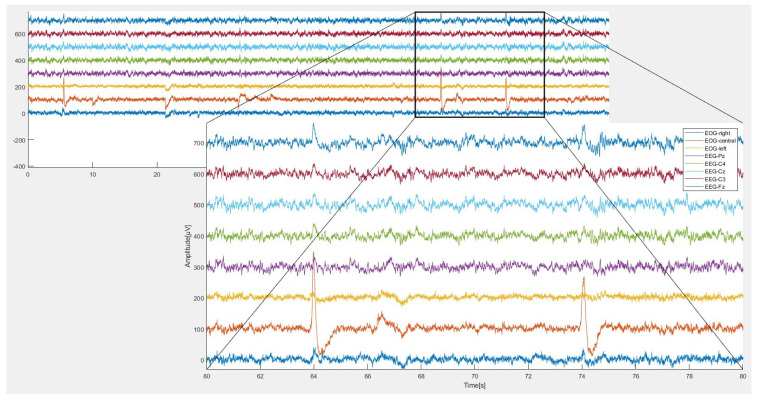
The original EEG with 22 channels of the MI-based BCI Competition IV dataset 2a.

**Figure 5 sensors-23-03889-f005:**
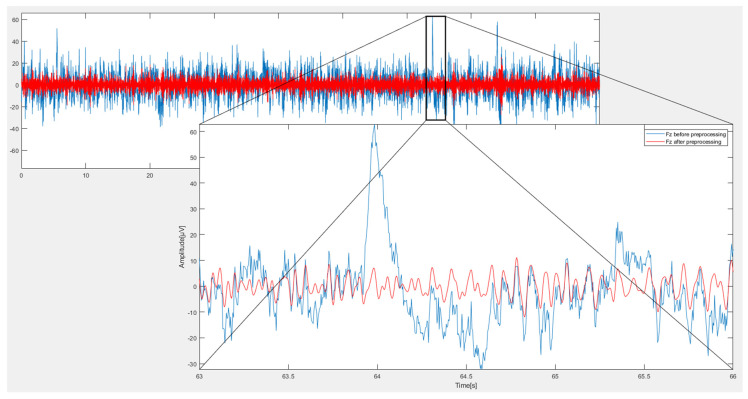
Fz channel before and after using preprocessing stage.

**Figure 6 sensors-23-03889-f006:**
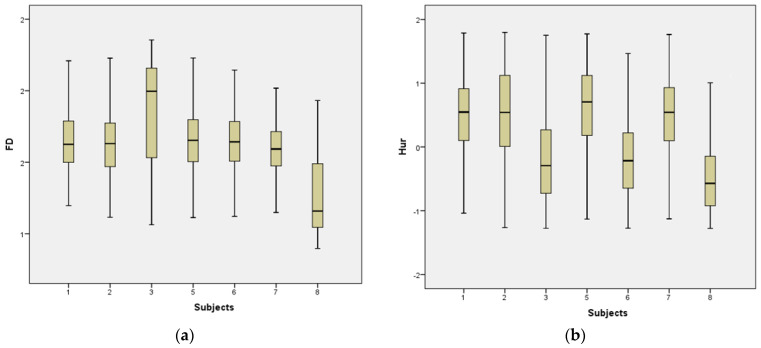
(**a**) The FD distribution of features and (**b**) the Hur distribution of features for each of the eight people who took part in the EEG MI-based BCI dataset.

**Figure 7 sensors-23-03889-f007:**
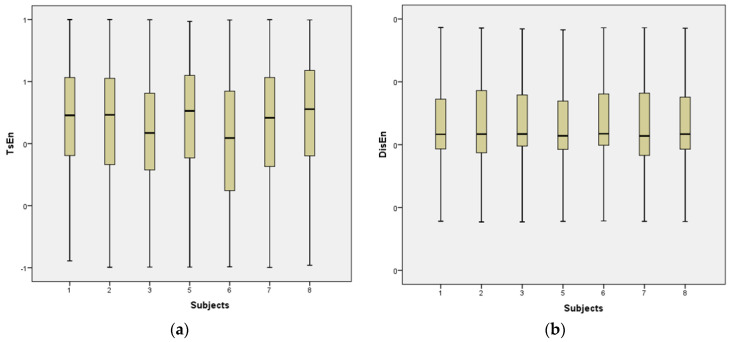
(**a**) The TsEn distribution of features and (**b**) the DispEn distribution of features for each of the eight people who took part in the EEG MI-based BCI dataset.

**Figure 8 sensors-23-03889-f008:**
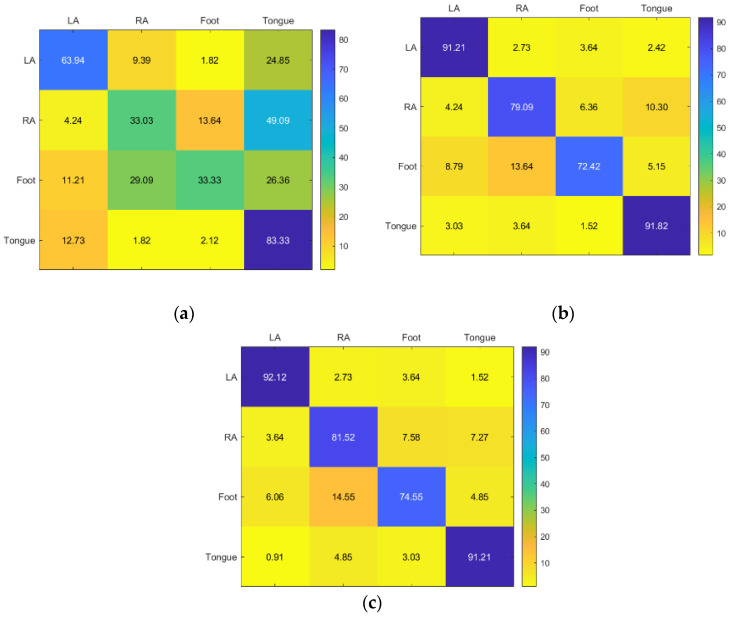
Confusion matrices for the (**a**) KNN, (**b**) SVM, and (**c**) RF classifiers to demonstrate the MI-BCI performance from the first subject’s four classes (left hand, right hand, feet, and tongue).

**Figure 9 sensors-23-03889-f009:**
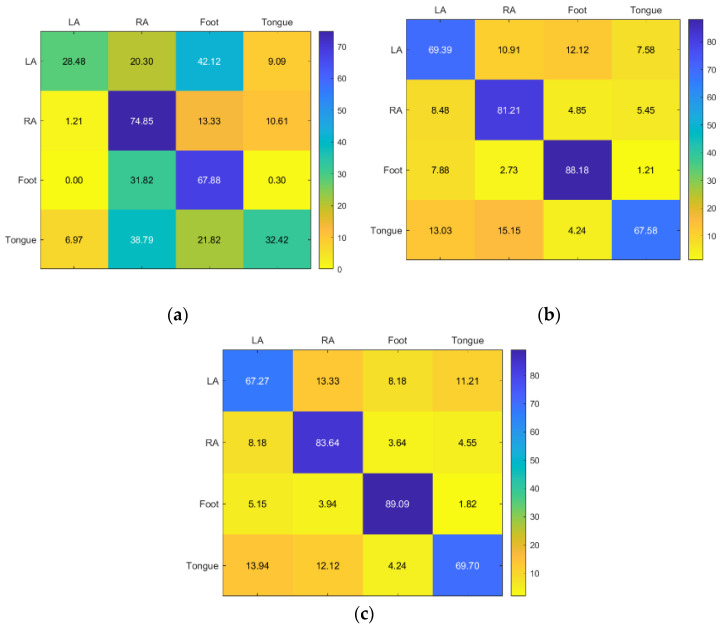
Confusion matrices for the (**a**) KNN, (**b**) SVM, and (**c**) RF classifiers to demonstrate the MI-BCI performance from the second subject’s four classes (left hand, right hand, feet, and tongue).

**Figure 10 sensors-23-03889-f010:**
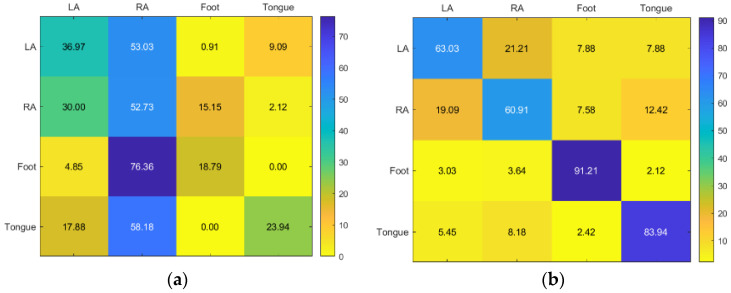
Confusion matrices for the (**a**) KNN, (**b**) SVM, and (**c**) RF classifiers to demonstrate the MI-BCI performance from the third subject’s four classes (left hand, right hand, feet, and tongue).

**Figure 11 sensors-23-03889-f011:**
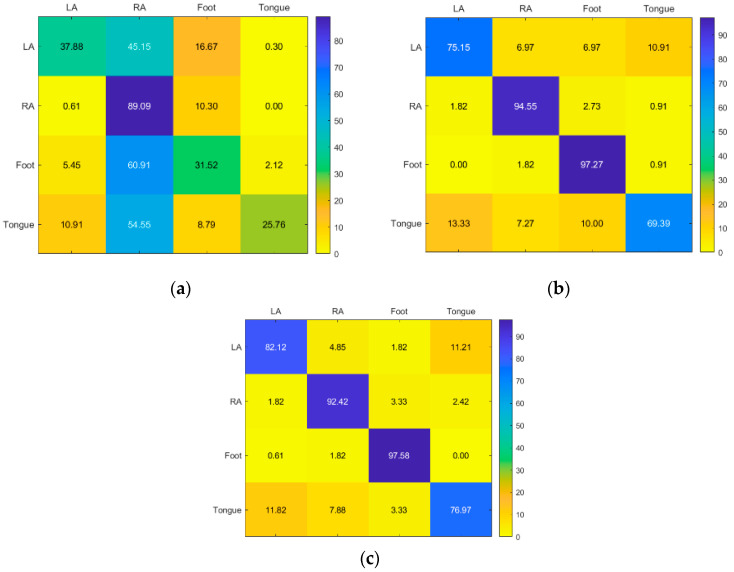
Confusion matrices for the (**a**) KNN, (**b**) SVM, and (**c**) RF classifiers to demonstrate the MI-BCI performance from the fifth subject’s four classes (left hand, right hand, feet, and tongue).

**Figure 12 sensors-23-03889-f012:**
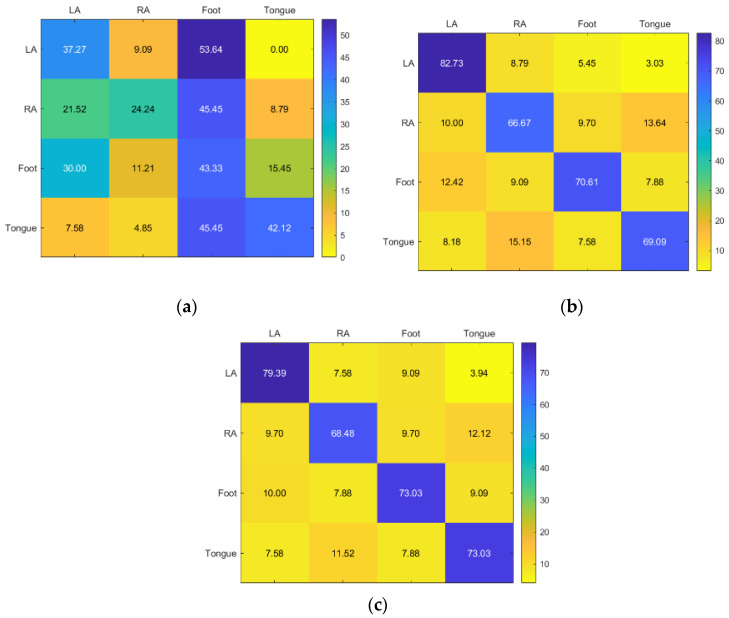
Confusion matrices for the (**a**) KNN, (**b**) SVM, and (**c**) RF classifiers to demonstrate the MI-BCI performance from the sixth subject’s four classes (left hand, right hand, feet, and tongue).

**Figure 13 sensors-23-03889-f013:**
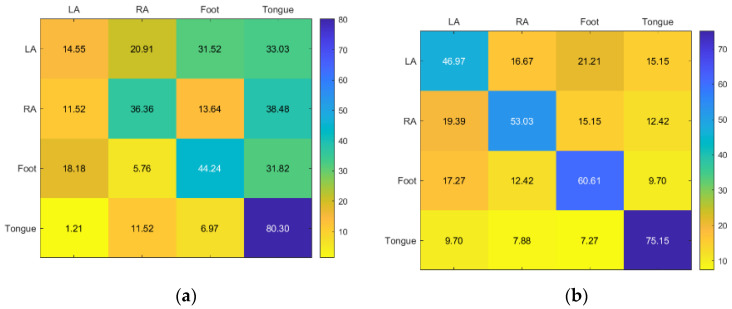
Confusion matrices for the (**a**) KNN, (**b**) SVM, and (**c**) RF classifiers to demonstrate the MI-BCI performance from the seventh subject’s four classes (left hand, right hand, feet, and tongue).

**Figure 14 sensors-23-03889-f014:**
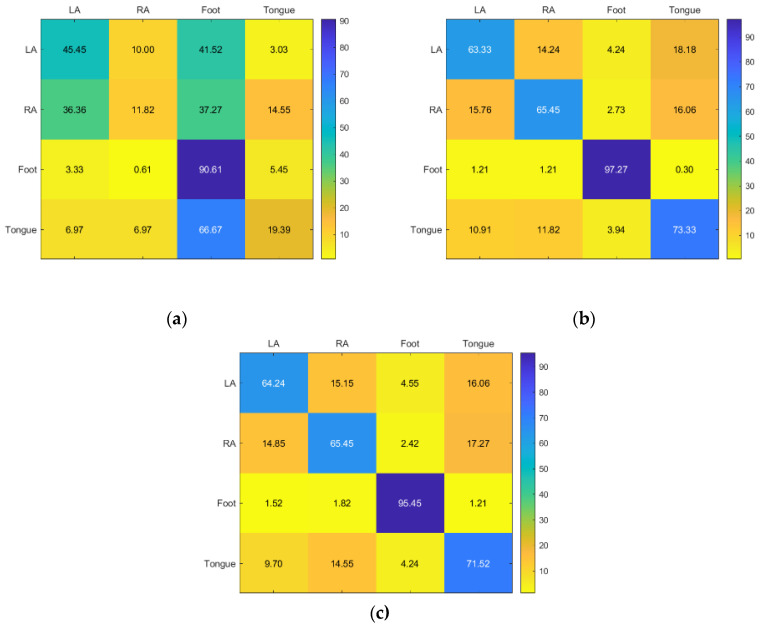
Confusion matrices for the (**a**) KNN, (**b**) SVM, and (**c**) RF classifiers to demonstrate the MI-BCI performance from the eighth subject’s four classes (left hand, right hand, feet, and tongue).

**Figure 15 sensors-23-03889-f015:**
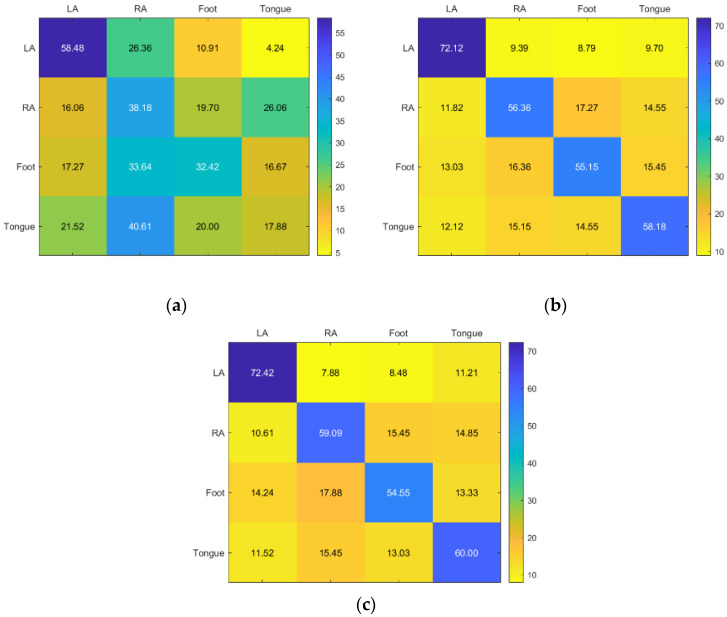
Confusion matrices for the (**a**) KNN, (**b**) SVM, and (**c**) RF classifiers to demonstrate the MI-BCI performance from the ninth subject’s four classes (left hand, right hand, feet, and tongue).

**Table 1 sensors-23-03889-t001:** State-of-the-art: various denoising, feature extraction, and classification approaches are used in EEG analysis for MI-based BCI.

Study	Denoising Technique	Feature Extraction	Classifiers
Liu et al. [33]	Bandpass filter (0.5–100) Hz, notch filter	CSP	SVM, KNN
Krishna et al. [34]	Moving average filter, band pass filter	Cross-correlation	SVM, KNN, LDA, NB, DT
Rejer et al. [36]	FastICA algorithm	Power band	SVM
Assi et al. [37]	Temporal filtering, spatial filtering, K means-ICA	Band power, DWT-band power, DWT-coherence, DWT-PLV	LDA and SVM
Selim et al. [17]	Butterworth filter	CSP	SVM
Ghumman et al. [38]	ICA	CSP	SVM
Narayan et al. [35]	Butterworth filter (8 to 30) Hz, notch filter, ICA	CSP, PCA	SVM, LDA
Al-Qazzaz et al. [5]	Conventional filtering, AICA WT denoising technique	Time domain, frequency domain, entropy domain	SVM, KNN, RF

**Table 2 sensors-23-03889-t002:** Sociodemographic information is displayed for the subjects with SAQ scores, (age in years, SAQ mean ± standard deviation SD).

Subject ID	Sex	Age	Handedness
S1	Female	22	Right
S2	Female	24	Right
S3	Male	26	Right
S5	Male	24	Right
S6	Female	23	Right
S7	Male	25	Left
S8	Male	23	Right
S9	Male	17	Right

**Table 3 sensors-23-03889-t003:** The average values (mean ± SD) of Hur to estimate the MI-BCI performance from four classes (left hand, right hand, feet, and tongue) of eight subjects individually.

Hur	LA	RA	Foot	Tongue	p Value
S1	0.863 ± 0.045	0.86 ± 0.054	0.874 ± 0.041	0.854 ± 0.04	0.05
S2	0.815 ± 0.061	0.894 ± 0.058	0.899 ± 0.056	0.82 ± 0.054	0.05
S3	0.843 ± 0.069	0.792 ± 0.049	0.785 ± 0.061	0.795 ± 0.057	0.05
S5	0.894 ± 0.043	0.873 ± 0.044	0.847 ± 0.065	0.872 ± 0.055	0.05
S6	0.775 ± 0.051	0.806 ± 0.045	0.808 ± 0.06	0.794 ± 0.049	0.05
S7	0.879 ± 0.048	0.835 ± 0.046	0.872 ± 0.045	0.877 ± 0.046	0.05
S8	0.807 ± 0.056	0.757 ± 0.052	0.777 ± 0.055	0.719 ± 0.051	0.05
S9	0.739 ± 0.054	0.758 ± 0.049	0.768 ± 0.068	0.775 ± 0.054	0.05

**Table 4 sensors-23-03889-t004:** The average values (mean ± SD) of FD to estimate the MI-BCI performance from four classes (left hand, right hand, feet, and tongue) of eight subjects individually.

FD	LA	RA	Foot	Tongue	p Value
S1	1.658 ± 0.067	1.654 ± 0.045	1.728 ± 0.077	1.575 ± 0.042	0.05
S2	1.569 ± 0.071	1.758 ± 0.065	1.679 ± 0.071	1.6 ± 0.088	0.05
S3	1.635 ± 0.082	1.575 ± 0.072	1.881 ± 0.039	1.852 ± 0.05	0.05
S5	1.714 ± 0.075	1.694 ± 0.073	1.682 ± 0.116	1.621 ± 0.085	0.05
S6	1.647 ± 0.079	1.668 ± 0.088	1.635 ± 0.076	1.649 ± 0.057	0.05
S7	1.59 ± 0.095	1.65 ± 0.069	1.606 ± 0.053	1.671 ± 0.07	0.05
S8	1.473 ± 0.061	1.388 ± 0.065	1.374 ± 0.053	1.643 ± 0.083	0.05
S9	1.65 ± 0.086	1.447 ± 0.098	1.607 ± 0.063	1.588 ± 0.098	0.05

**Table 5 sensors-23-03889-t005:** The average values (mean ± SD) of TsEn to estimate the MI-BCI performance from four classes (left hand, right hand, feet, and tongue) of eight subjects individually.

TsEn	LA	RA	Foot	Tongue	p Value
S1	3.096 ± 0.181	3.169 ± 0.112	3.15 ± 0.097	3.173 ± 0.078	0.05
S2	3.169 ± 0.088	3.164 ± 0.155	3.115 ± 0.174	3.153 ± 0.12	0.05
S3	3.153 ± 0.101	3.205 ± 0.071	3.066 ± 0.179	3.112 ± 0.082	0.05
S5	3.134 ± 0.099	3.105 ± 0.192	3.1 ± 0.251	3.169 ± 0.114	0.05
S6	3.048 ± 0.159	3.132 ± 0.078	3.107 ± 0.113	3.093 ± 0.286	0.05
S7	3.198 ± 0.102	3.156 ± 0.067	3.151 ± 0.112	3.12 ± 0.111	0.05
S8	3.151 ± 0.098	3.193 ± 0.097	3.091 ± 0.418	3.129 ± 0.154	0.05
S9	3.078 ± 0.14	3.157 ± 0.095	3.117 ± 0.116	3.159 ± 0.089	0.05

**Table 6 sensors-23-03889-t006:** The average values (mean ± SD) of DispEn to estimate the MI-BCI performance from four classes (left hand, right hand, feet, and tongue) of eight subjects individually.

DispEn	LA	RA	Foot	Tongue	p Value
S1	1.382 ± 0.012	1.379 ± 0.028	1.385 ± 0.003	1.385 ± 0.001	0.05
S2	1.385 ± 0.001	1.382 ± 0.017	1.38 ± 0.025	1.385 ± 0.003	0.05
S3	1.384 ± 0.003	1.385 ± 0.001	1.385 ± 0.001	1.385 ± 0.001	0.05
S5	1.385 ± 0.002	1.383 ± 0.004	1.383 ± 0.009	1.385 ± 0.002	0.05
S6	1.382 ± 0.009	1.385 ± 0.001	1.385 ± 0.002	1.377 ± 0.033	0.05
S7	1.385 ± 0.003	1.385 ± 0.001	1.384 ± 0.004	1.384 ± 0.006	0.075
S8	1.384 ± 0.003	1.385 ± 0.003	1.348 ± 0.161	1.385 ± 0.002	0.05
S9	1.383 ± 0.006	1.384 ± 0.003	1.385 ± 0.002	1.385 ± 0.002	0.05

**Table 7 sensors-23-03889-t007:** Performance comparison of SVM, KNN, and RF classifiers from four classes (left hand, right hand, feet, and tongue) of eight subjects individually.

Subjects	SVM Accuracy %	KNN Accuracy %	RF Accuracy %	SVM Precision %	KNN Precision %	RF Precision %	SVM Recall %	KNN Recall %	RF Recall %
S1	53.4	83.63	84.84	56.33	53.41	83.71	83.63	76.49	76.82
S2	50.9	76.59	77.42	57.86	50.91	76.82	76.59	77.43	77.42
S3	33.1	74.77	75.68	46.3	33.11	74.23	74.77	75.47	75.68
S5	46.06	84.09	87.27	60.75	46.06	84.09	84.09	87.14	87.27
S6	36.74	72.27	73.48	43.57	36.74	72.33	72.27	73.45	73.48
S7	43.86	58.93	61.51	42.6	43.86	58.56	58.94	61.26	61.52
S8	41.81	74.84	74.16	43.41	41.82	74.48	74.85	73.89	74.16
S9	36.74	60.45	61.51	36.43	36.74	60.27	60.45	61.37	61.52
Average	42.83	73.2	74.48	48.41	42.83	73.06	73.2	73.31	73.48

## Data Availability

http://www.bbci.de/competition/iV (Accessed on 17 October 2020).

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
