# Peer review of "EEG Signal Complexity Measurements to Enhance BCI-Based Stroke Patients’ Rehabilitation"

_sensors, 2023, doi:10.3390/s23083889_

Round 1

Reviewer 1 Report

Authors proposed a title " EEG Signal Complexity Measurements to Enhance
BCI- Based Stroke Patients' Rehabilitation".After carefully reading manuscript,there are certain queries need to be justified from authors and accordingly manuscript need to be modified.

1. In Title,authors specifically mentioned about EEG signal complexity measurement. How this particular terminology addressed in manuscript.

2. There are so many dimensionality reduction or feature selection algorithms are available ? Why authors have chosen Laplacian Eigenmap ? Kindly justify.

3. Which features are extracted from EEG signals,kindly include detail in tabular form.Further it is always beneficial to visualize the distribution of features.In recent paper,authors have utilized Box plot for feature distribution.Kindly refer following paper and add in revised manuscript with suitable discussion.

a. https://www.sciencedirect.com/science/article/abs/pii/S0375960121006642

4. It is always a better practice to apply 10-fold cross validation for classification/regression based prediction results.It is neccessary,since training and testing results are biased.For more idea,kindly refer 3a.

5. Suggested to add numerical values in introduction as well as conclusion section.

6. Authors need to highlight specific contribution as compared to published literature.

7. Challenges and future direction should also be added in revised version, so that readers can better understand author's proposed methodology.

Author Response

We have carefully revised the manuscript following the Editorial office and Reviewers’ comments. We considered and addressed each one of their concerns and remarks.

We are grateful for the feedback provided by the Editor and Reviewers. Their remarks and suggestions helped us to improve the manuscript. We hope that the revised version of the study has addressed all your concerns and will be considered a contribution of interest to the readership of “Sensors Journal.”

For your convenience, a list of responses to the Reviewers’ remarks is the attached file.

Reviewer 2 Report

Despite the paper is well written and the methodology is well motivated, there are some major issues to be solved before considering for publication.

1.      What is the gap in the studies in the literature? What is the main contribution of the present work?

2.      Please include clinical significance of the proposed study.

3.      How the proposed system would be useful in the real-world scenario? Need to discuss.

4.      Include the limitations and future directions of this work.

5.      What about computational complexity of the proposed method? Some description related to computational complexity is required in the paper.

6.      The proposed approach was evaluated using 8 subjects EEG. I strongly suggest the authors to recruit and collect date from more subjects to strengthen the findings. Otherwise, this work is just another incremental addition.

7.      The authors used band pass filter with the cutoff frequency of 8-30 Hz. What was the purpose notch filter here?

8.      In recent days, deep learning algorithms outperforms in many research fields. I am wondering why the authors didn’t use deep learning algorithms. Need to discuss.

9.      The authors never state what level of performance would be considered adequate for a “real-world scenario” system that is clinically useful and valid. 

10.  Please, discuss whether the application of the algorithm is possible for online detection (i.e., close loop system).

11.  What are the strengths and weaknesses of the proposed framework? Please include them in the paper.

Author Response

Editorial Office and Reviewer’s Comments and Authors’ Responses

Noor Kamal Al-Qazzaz, Alaa A. Aldoori, Sawal Hamid Bin Mohd Ali, Siti Anom Ahmad, Ahmed Kazem Mohammed, and Mustafa Ibrahim Mohyee

Sensors Journal ID: sensors-2230962

EEG Signal Complexity Measurements to Enhance BCI- Based Stroke Patients' Rehabilitation

Editorial Office’s Comments and Authors’ Response

Editorial Office’s Comments

Major Revisions 

Authors’ Response

We have carefully revised the manuscript following the Editorial office and Reviewers’ comments. We considered and addressed each one of their concerns and remarks.

We are grateful for the feedback provided by the Editor and Reviewers. Their remarks and suggestions helped us to improve the manuscript. We hope that the revised version of the study has addressed all your concerns and will be considered a contribution of interest to the readership of “Sensors Journal.”

For your convenience, a list of responses to the Reviewers’ remarks is the attached file.

Reviewer 3 Report

- The introduction should focus more on articulating the motivation and contributions. 

- Section 3, Figure 1 should receive more elaboration and what does it mean it shows the directions!!!

- The manuscript mentions that the data is for 8 subjects, is this enough to make a proper statistical analysis?

- Sections 3.2,3.3 includes a alot of theoretical background information that is not part of the contribution of this manuscript. This is especially true becuase some parts show great similarity to the word with Internet sources. The manuscript should be more succinct and crisp to serve the conducted work properly. 

- The numbers inside the confusion matrices need to be made larger for good readership. 

- Define the performance paramters and include other values (e.g., precision)

- Line 87, what is Mabrouk? I udnerstand that it is a study by a certain author, but proper citations should be included (not at the end in this case).

- The table of abbreviations is missing but required by the journal template. 

- Some references are laking proper complete information (e.g., 18 and 21)

- Similar studies utilizing EEG and relating to brain activity and emotions should be discussed, see Fraiwan, M., Alafeef, M. & Almomani, F. Gauging human visual interest using multiscale entropy analysis of EEG signals. J Ambient Intell Human Comput 12, 2435–2447 (2021). https://doi.org/10.1007/s12652-020-02381-5

Alafeef, M., Fraiwan, M. On the diagnosis of idiopathic Parkinson’s disease using continuous wavelet transform complex plot. J Ambient Intell Human Comput 10, 2805–2815 (2019). https://doi.org/10.1007/s12652-018-1014-x

Author Response

(The authors gave the same response as above.)

Round 2

Reviewer 1 Report

Authors have addressed all reviewer comments with justifications and revised manuscript accordingly.

Reviewer 3 Report

The authors addressed my comments.